# Characterization of Lemon Pepper and Black Ginger Extracts and Macroemulsions as Natural Pain Relievers for Spice Stick Balsam Formulation

**DOI:** 10.3390/ph16030371

**Published:** 2023-03-01

**Authors:** Celinia Harijono, Bibiana Widiyati Lay

**Affiliations:** 1Faculty of Biotechnology, Atma Jaya Catholic University of Indonesia, Jakarta 12930, Indonesia; 2Research Center for Indonesian Spices, Atma Jaya Catholic University of Indonesia, Jakarta 12930, Indonesia

**Keywords:** lemon pepper, black ginger, macroemulsions, antioxidant activity, spicy stick balsam

## Abstract

Lemon pepper or andaliman (*Zanthoxylum acanthopodium*) and black ginger (*Kaempferia parviflora*) are rich in bioactive compounds that possess antioxidant and anti-inflammatory activities. Our recent study demonstrated that andaliman ethanolic extract also exerted anti-arthritic and anti-inflammatory effects in arthritic mice in vivo. Therefore, natural anti-inflammatory and anti-arthritic compounds for alternative natural pain relievers in balsam formulation are needed. This study aimed to produce and characterize lemon pepper and black ginger extracts and their macroemulsion products, followed by formulation, characterization, and stability of spice stick balsam products containing lemon pepper and black ginger macroemulsions. The extraction yields obtained were 24% *w*/*w* for lemon pepper and 59% *w*/*w* for black ginger. GC/MS results showed that the lemon pepper extract contained limonene and geraniol compounds, and black ginger extract contained gingerol, shogaol, and tetramethoxyflavone compounds. Spice extracts were successfully made in the form of a stable emulsion. The antioxidant activity in both spice extracts and emulsions was relatively high (>50%). The five stick balsam formulas obtained had a pH of 5, 4.5–4.8 cm spread ability, and 30–50 s of adhesion. The stability of products showed no microbial contamination. Based on the organoleptic results, the stick balsam formula of black ginger and black ginger: lemon pepper (1:3) was the most preferred by the panelists. In conclusion, lemon pepper and black ginger extracts and macroemulsions could be used as natural pain relievers in stick balsam products to promote health protection.

## 1. Introduction

Arthritis is defined as an acute or chronic joint inflammation that often co-exists with pain and structural damage. In Indonesia, the prevalence of arthritis is 7.3% and continues to increase. Treatment for arthritis is focused on reducing pain and inflammation in a joint and improving quality of life by using medication such as oral and topical drugs. Balsam is popularly known as one of the topical and over-the-counter (OTC) medications that has functioned as a pain relief to reduce the inflammation. According to Indonesian Agency for Drug and Food Control, balsam belongs to quasi medicine containing certain active ingredients with pharmacological effects to treat mild symptoms [1]. In the USA, balsam is grouped as one of safe and effective nonprescription drugs for reducing pain. Balsam has a semi-solid texture and like an ointment that is directly applied to hands then the hands become hot and sticky. Balsam is commonly used by applying it to the painful area and providing a relaxing warm feeling [2]. However, balsam cannot completely cure arthritis, it can only reduce the pain. In general, balsam contains an active ingredient called methyl salicylate, but this ingredient has limitation to cause allergies such as redness, itching, and burning sensations [3]. To solve this problem, searching and exploration for alternative natural compounds with anti-inflammatory efficacy become the focus of investigation. Indonesia is known as one of the most biodiversity-rich countries in the world with its high medicinal plant collection including spices and herbs.

Lemon pepper (*Zanthoxylum acanthopodium*; Rutaceae), known as andaliman in Indonesian, is a spice plant that grows in North Sumatra. It is often used as a flavoring agent in Batak cuisine, with a bitter taste and a lemon-like aroma. Lemon pepper contains a variety of secondary metabolites such as phenols, saponins, flavonoids, tannins, triterpenoids, and steroids. Limonene in lemon pepper is known to have antioxidant and anti- inflammatory activity [4]. Our previous study showed that andaliman ethanolic extract possessed potential anti-inflammatory and xanthin oxidase inhibitory activities [5]. A recent report by Setiadi et al. demonstrated that andaliman ethanolic extract and its nanoandaliman extract exerted potential anti-arthritic and anti-inflammatory activities in inflammatory arthritic mice model in vivo [6]. Here, andaliman extract was further applied as an alternative topical medicine to reduce painful inflammation. Black ginger (*Kaempferia parviflora*; Zingiberaceae) known as kra-chai-dam is a spice plant from Thailand. It is often used as a tonic and traditional medicine, with a distinctive spicy and bitter taste. Black ginger is known to be rich in secondary metabolites such as methoxyflavones, phenolic glycosides, polyphenols, and terpenoids. Methoxyflavones in black ginger are known to have antioxidant and anti-inflammatory activity [7]. Therefore, these two spices were chosen as the main ingredients for making stick balsam. This study aimed to produce and characterize lemon pepper (LP) and black ginger (BG) extracts and their microemulsion products, including physicochemical parameters, stability, and antioxidant activity as natural pain reliever candidates. Further, these LP and BG macroemulsions were applied for formulation, characterization, and stability of spice stick balsam products.

## 2. Results

### 2.1. Yield and Active Compounds Profiling of Lemon Pepper and Black Ginger Extracts

The BG was dried using the freeze-drying method and obtained a yield of 33.45% ± 0.01 *w*/*w*. Extractions of LP and BG were carried out by the mechanical maceration method using food grade ethanol 70%. The yields obtained were 24% ± 0.00 *w*/*w* for LP and 59% ± 0.02 *w*/*w* for BG. The bioactive compounds profiling results in LP and BG were shown in Figure 1. Bioactive compounds with potential anti-inflammatory agents in both spices were identified and categorized (Table 1). Based on the results, five compounds that have the potential as anti-inflammatory agents in LP and BG were found.

### 2.2. Characteristics of Lemon Pepper and Black Ginger Macroemulsions

The emulsion from LP had a brownish color with a strong scent. The emulsion of BG had a purple-black color with a fresh scent. Both spice emulsions did not solidify at room temperature. The particle size of the emulsions was measured with a particle size analyzer. The particle diameter of the LP emulsion was 10,432.4 nm, and the BG emulsion was 8286.7 nm.

### 2.3. Antioxidant Activity of Lemon Pepper and Black Ginger Extracts and Macroemulsions

Figure 2 showed various antioxidant activities of the LP and BG extracts and emulsions. The LP extract had higher antioxidant activity (93.04% ± 0.56) than the BG extract (64.47% ± 2.27). All emulsions had high antioxidant activity (>50%), but when compared to extracts, the antioxidant activity of emulsions tended to be lower.

### 2.4. Characteristics and Stability of Lemon Pepper and Black Ginger Stick Balsams

Based on the visual, the stick balsams had a compact texture and were not too oily, and also had a yellowish color in the LP formula, a purplish color in BG and BG:LP (3:1) formulas, and a brownish color in BG:LP (1:1 and 1:3) formulas (Figure 3). The scent was quite refreshing and tended to have a menthol scent. All formulas are declared homogeneous, as seen from their color and the absence of granules in the products. The spread ability of the product was in the 4.5–4.8 cm range, the product adhesion was in the 30–50 s range, and the pH was 5 for each spice balsam (Table 2). According to the results of product stability, no bacterial colonies were found growing on the media from day 0, 7, 14, and 30 until day 60. All stick balsam products had TPC value of 0 CFU/g.

### 2.5. Organoleptic Results of Lemon Pepper and Black Ginger Stick Balsams

The organoleptic results showed hedonic data from 25 untrained panelists on the color, aroma, texture, warmth, and overall value of each stick balsam product (Table 3). The panelists’ average preference for the five parameters was high (>5). The color, aroma, and texture parameters were outperformed by control stick balsam and followed by BG stick balsam for color parameters and LP stick balsam for aroma and texture parameters. On warmth parameters, panelists tended to prefer the BG:LP (1:3) stick balsam. The most preferred formulations were the BG stick balsam and the combination of BG and LP (1:3) stick balsam based on the overall value (Figure 4).

## 3. Discussion

In this study, LP and BG were made into extracts to obtain the bioactive compounds. The yields from LP and BG extracts with 70% food-grade ethanol were 24% ± 0.00 *w*/*w* and 59% ± 0.02 *w*/*w*, respectively. The extracts were identified using GC/MS to determine the bioactive compounds. The results showed that monoterpene compounds were found in the LP extract, as well as phenolic, polyphenols, methoxyflavones, and xanthone compounds in the BG extract (Figure 1 and Table 1). Monoterpene compounds found in LP extract were known to have anti-inflammatory activity that can be developed to treat arthritis symptoms [4,8]. Phenolic and polyphenol compounds such as shogaol, iso-shogaol, and gingerol were commonly found in ginger rhizomes except for BG [9]. Interestingly in this study, they were found in BG (Table 1). These compounds give ginger a warm and spicy taste [10]. Methoxyflavone compounds were not found in other types of ginger except for BG. This compound was known to have anti-inflammatory activity that can relieve gout and osteoarthritis [7]. The xanthone compound in BG gives the blackish purple pigment to the ginger rhizome. This compound was known as an anti-inflammatory agent and could be used as a natural dye [11]. The LP and BG extracts were then made into an emulsion to facilitate the homogenization in the stick balsam-making process. The results showed that the emulsions made were stable at room temperature and did not solidify due to the soy oil used in the production of emulsions having a low freezing point of −20°C [12]. LP and BG emulsions could mix with the balsam-making oil homogeneously. The particle size of LP and BG emulsions were measured by PSA and the results were 10,432.4 nm for LP emulsion and 8286.7 nm for BG emulsion. Based on the particle size, both emulsions were categorized into macroemulsions (>100 nm) [13].

Antioxidants are useful to ward off free radicals. In topical products, antioxidants can help treat and protect the skin. An antioxidant activity assay was needed to determine the efficacy of LP and BG in the form of extracts and emulsions using DPPH as the reagent. DPPH is a free radical compound that has a high absorption at 517 nm wavelength with a purple solution color. DPPH is an unstable compound that requires an electron or H radical donor to be a stable non- radical compound [14]. Monoterpene in LP, as well as gingerol, shogaol, and tetramethoxyflavone in BG, were known as antioxidant compounds that can be an electron or H atom donors to stabilize DPPH compounds [7,8,10]. The results showed that both spice extracts and emulsions had high antioxidant activity (Figure 2). Formulation of emulsions from the extracts could significantly reduce the antioxidant activity, due to the addition of soybean oil and tween 80 which can inhibit the antioxidant activity of the extracts [15]. In the combination emulsion, especially the combination of BG and LP 1:3 and 3:1, there were no significant differences with the antioxidant activity of BG extract. In addition, the combination emulsions had antioxidant activity values that were close to the antioxidant activity values of both extracts. It can be concluded that the best antioxidant activity in the emulsion was found in the emulsion made from a combination of BG and LP.

The spice stick balsam product was successfully made in as many as 5 formulas. Based on Table 2, the spice stick balsams have met the good characteristics of the balsam product according to Anastasia and Romadhonni on the parameters of homogeneity, adhesion, and pH [2]. Good adhesion of the balsam is >4 s, which means that the balsam can adhere perfectly to the skin and cannot be easily rinsed off. In addition, the stick balsam products did not leave an excessively sticky feeling on the skin. Based on the pH of the product, it was known that the product is safe to use on the skin because its pH is in the skin’s pH range of 4.5–6.5. On the spread ability parameter, it did not meet the criteria for a good balsam preparation. However, when compared with the characteristics of the commercial stick balsam, the spice balsams have similar characteristics to the commercial product. In addition, our results also demonstrated that the stick balsam products were stable against the growth of contaminant microorganisms with a storage period of 2 months at room temperature, due to zero microbial contamination. All stick balsam products met the total plate count requirements of the Indonesian Agency for Drug and Food Control. This regulation stated that the plate count number for each product was less than 10^7^ CFU/g [16]. The BHT preservative added to the product can effectively inhibit bacterial growth. In addition, it is also known that the extracts of LP and BG have potential compounds as natural preservatives. Essential oil monoterpenoid compounds in LP, as well as tetramethoxyflavone and mangostine compounds in BG have antimicrobial activity that can inhibit the growth of microorganisms in the product [4,7,11].

The hedonic test was used to determine the preference of panelists for a product. The examination was conducted to assess the opinion of the panelists on the newly developed product [17]. The hedonic results showed that the majority of the panelists preferred the stick balsam formulas with BG and BG:LP (1:3) combination (Figure 4). Based on the statistical results (Table 3), no significant difference was found between the spice stick balsam and the control. Therefore, it could be concluded that these spice stick balsams have the potential to compete with commercial products.

## 4. Materials and Methods

### 4.1. Spices Preparation and Extraction

LP was obtained as dried fruits from a traditional market in Balige (North Sumatera province, Indonesia), while BG was obtained in fresh form from a modern market in Tangerang (Banten province, Indonesia). The BG was washed and sliced thinly, and then frozen in the freezer for 24 h before being put in the freeze-dryer. Freeze-drying of BG was carried out at a temperature of −11 °C and pressure of 1 atm for 10 days. Both spices were grounded with a food processor separately, then sieved through a 100 × mesh sieve. The spices powder was stored in a standing pouch at room temperature. Both spices were extracted using the mechanical maceration method based on Pangabean et al. for LP and Rahman et al. for BG with modifications [18,19]. Powdered spices were macerated with 70% food-grade ethanol with a sample and solvent ratio of 10% (*w*/*v*) for LP and 4% (*w*/*v*) for BG. Samples were left for 180 min in a water bath shaker at a speed of 70 rpm with a temperature of 55 °C for LP and 75 °C for BG. After that, the solutions were filtered with filter paper to obtain the filtrate. The filtrate was evaporated using a rotary evaporator at a temperature of 55 °C, speed of 60 rpm, and pressure of 90 mbar to evaporate the solvent in the filtrate. The remaining solvents were evaporated using Petri dishes and dried in a fume hood. The thick extract was kept in a conical tube in the refrigerator. The yield of extract was calculated with the following formula: Yield (%) = [Final dry weight/Initial dry weight] × 100%.

### 4.2. Active Compounds Profiling

Bioactive compounds from LP and BG were identified using gas chromatography-mass spectrometry (GC/MS) based on the method of Devi et al. for LP and Pitakpawasutthi et al. for BG with modification [20,21]. The spice extract was dissolved in 70% ethanol at a ratio of 10% *w*/*v*, then filtered through a membrane filter. The GC/MS used is equipped with a TG5MS column and helium gas with a flow rate of 5.4 mL/minute for LP and 1.0 mL/minute for BG. The injected sample volume was 1 µL. Identification of LP bioactive compounds was conducted at 40 °C oven temperature for 5 min and an increase in temperature of 10 °C/minute until the temperature reached 230 °C and maintained for 2 min. For BG, the identification was conducted at 60 °C oven temperature for 1 min and the temperature was increased by 3 °C/minute until the temperature reached 240 °C. Mass spectrometry was calculated in EI mode with a voltage of 70 eV. The mass spectrometry results were compared with the database provided by the NIST_MS software.

### 4.3. Macroemulsion Formulation

The emulsion was made using Hayati and Balqis method with modification [22]. The emulsion was made by making solution A (tween 80, aquadest, and 96% food-grade ethanol) and solution B (soy oil, spice extract, and propylene glycol) separately according to Table 4. Solution A was made by heating the tween 80 and aquadest until both substances mixed homogenously, then 96% food-grade ethanol was added after the solution had cooled enough. Both solutions were mixed and stirred homogeneously, then stored in a conical tube at room temperature. The particle size of emulsions was determined at Nanotech Herbal Indonesia, Tangerang (Indonesia) by a particle size analyzer.

### 4.4. Antioxidant Activity Assay

The antioxidant activity assay was conducted by the method of Ghimeray et al. with modifications [23]. The antioxidant activity of LP and BG extracts and their macroemulsion products was tested using 0.6 mM 2,2-diphenyl-1-picryl-hydrazyl- hydrate (DPPH) solution in methanol. A total of 100 µL of 0.5% sample solution in methanol was mixed with 100 µL of DPPH solution. In this test, the control used was methanol mixed with DPPH solution, while the reference used was 0.1% ascorbic acid solution in methanol with DPPH solution. Each solution was vortexed and incubated at room temperature for 30 min. The absorbance measurement was conducted with a microplate reader at a wavelength of 517 nm. The assay was done with triplicate repetitions.

### 4.5. Spice Balsam Formulation

The formulation of stick balsam and the making process were conducted using the method of Athaillah and Lianda with modifications [24]. A total of 5 formulations of spice stick balsam and one control balsam were made according to Table 5 with the total weight of each stick balsam being 5 g. The oil base ingredients (paraffin, cera alba, and vaseline) were melted at a temperature of 60–70 °C, then the extract emulsion, butyl-hydroxytoluene (BHT), and menthol were added, then homogenized. The balsam mixture was immediately molded into the stick tube and stored at room temperature. For control balsam, it was similarly prepared with ingredients and composition of spice stick balsams without the addition of lemon pepper and black ginger macroemulsions.

### 4.6. Characteristic Tests

The characteristic tests were conducted based on the method of Anastasia and Romadhonni with modifications [2]. The tests were done in triplicate for homogeneity test, pH analysis, spread ability, and adhesion tests. In the homogeneity test, 0.5 g of balsam product was weighed and smeared on an object glass, then observed for texture and color of the product. The pH analysis was carried out by applying the balsam product directly to the pH indicator strips and observing the color change on the pH indicator. The spread ability and adhesion tests were carried out continuously. The petri dish was weighed and loaded until the total weight of 1 part of the Petri dish is 100 g. The balsam from each formulation was weighed as much as 0.5 g and placed on an inverted Petri dish. The Petri dish was pressed with its lid in an inverted position and was given a load of 50 g. The width of the sample dispersion was then measured. After being measured and observed, the lid of the Petri dish is slowly lifted, and the time it takes for the Petri dish to fall is calculated. The results were then compared with commercial stick balsam’s (GELIGA^®^, PT. Eagle Indo Pharma, Tangerang, Banten, Indonesia) characteristics.

### 4.7. Stability Test

The product stability test was conducted using the total plate count (TPC) method based on Abna et al. [25] with modifications, and the test was done in triplicate repetitions. The product was dissolved aseptically in a sterile 0.2 M phosphate buffer solution (pH 7) with 1% tween 20. The ratio of the sample to the solvent was 1:9 to get a 10^−1^ dilution, then diluted from a dilution of 10^−1^ to 10^−2^, then 10^−3^. The media used was tryptic soy agar media with the addition of 1% 2,3,5-triphenyltetrazolium chloride (1% concentration) sterile. Sample inoculation was done by the pour plate method. A total of 1 mL of samples from each dilution was put into a sterile Petri dish, then poured with agar media and homogenized. After the agar solidified, the test samples were incubated at 35 °C for 3 days. The number of microorganisms that grow on the media was counted and the value of microorganism contamination was determined. The desired total plate count number was <10^7^ CFU/g.

### 4.8. Organoleptic Test

The organoleptic test was conducted by the hedonic test method. The parameters tested were color, aroma, texture, warmth, and the overall value of the stick balsam product. These parameters were rated on a scale of 1–9 which represents the value of very much disliked to very much liked. The panelists used were 25 semi trained panelists from the Faculty of Biotechnology, Atma Jaya Catholic University of Indonesia. The obtained data were statistically tested, and the significant difference between samples in each hedonic parameter was determined.

### 4.9. Statistical Analysis

Antioxidant activity data and organoleptic data obtained were statistically processed using Microsoft Excel 2019 (Microsoft 365, Washington, DC, USA) and IBM SPSS Statistics 26.0 (IBM, New York, NY, USA). The standard deviation and the average values were determined using the descriptive statistics method. The significance values were determined from the data using the analysis of variance (ANOVA) with the Duncan method for the post hoc test of antioxidant activity data and Kruskal-Wallis with the stepwise stepdown method for the post hoc test of organoleptic data, with 95% confidence level (α = 0.05).

## 5. Conclusions

Spice extraction was conducted successfully with yields of 24% *w*/*w* for LP and 59% *w*/*w* for BG. The GC/MS results showed that the LP extract contained monoterpenoid compounds such as limonene and geraniol, and the BG extract contained gingerol, shogaol, and tetramethoxyflavone compounds which were known as anti-inflammatory agents as well as natural pain relievers. These spice extracts have been successfully made into macroemulsion forms that were stable at room temperature and can be used to formulate spice stick balsam products. Both extracts and emulsions had quite high antioxidant activity (>50%). Spice stick balsam has been successfully made in as many as 5 formulas, with a refreshing aroma and appearance that is quite dense but not too oily, similar to commercial stick balsam. The stick balsam products had good stability during a storage period of 2 months at room temperature. Based on panelists’ preferences, the most preferred stick balsam formulas were BG and BG:LP (1:3) by their overall value. In conclusion, the functional ingredients of LP and BG with anti-inflammatory, anti-arthritic, and antioxidant effects could be used as alternative natural pain relievers for developing topical pharmaceutical products such as spice stick balsams. Further study is needed to determine the effectiveness of stick balsam products in relieving the pain caused by arthritis through in vitro and in vivo approaches. In addition, allergy and skin irritation testing on the use of balsams is also needed to be performed.

## Figures and Tables

**Figure 1 pharmaceuticals-16-00371-f001:**
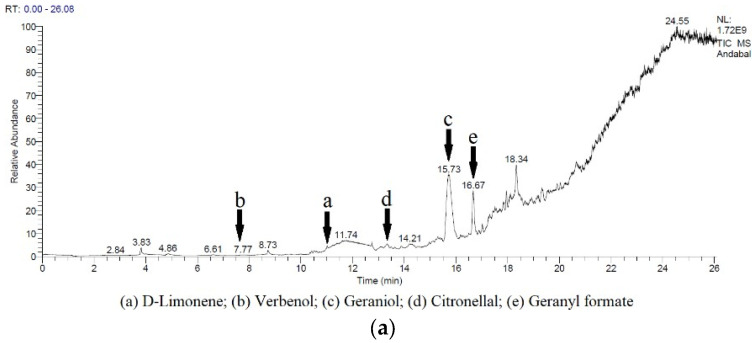
GC/MS chromatograms of lemon pepper (**a**) and black ginger (**b**) extracts.

**Figure 2 pharmaceuticals-16-00371-f002:**
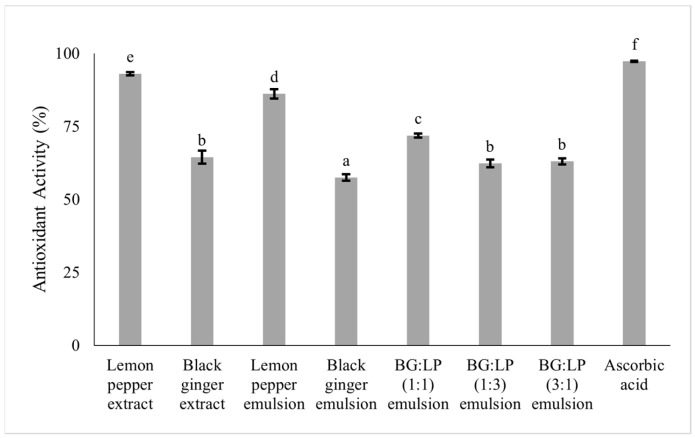
Antioxidant activity of lemon pepper (LP) and black ginger (BG) extracts and macroemulsions. Note: (a, b, c, d, e, f) letter differences indicate a significant difference between the samples at *p* < 0.05 according to Duncan method for post hoc test.

**Figure 3 pharmaceuticals-16-00371-f003:**
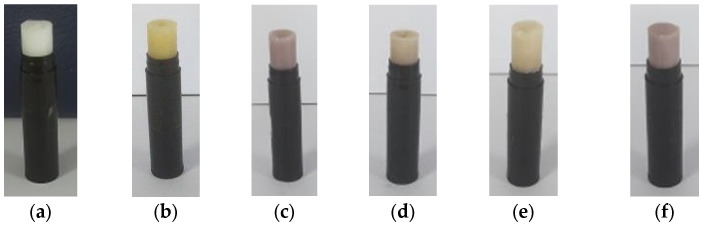
Spice stick balsams made from lemon pepper (LP) and black ginger (BG) macroemulsions. Note: Control (**a**); LP (**b**); BG (**c**); combination of BG:LP (1:1) (**d**); combination of BG:LP (1:3) (**e**); and BG:LP (3:1) (**f**).

**Figure 4 pharmaceuticals-16-00371-f004:**
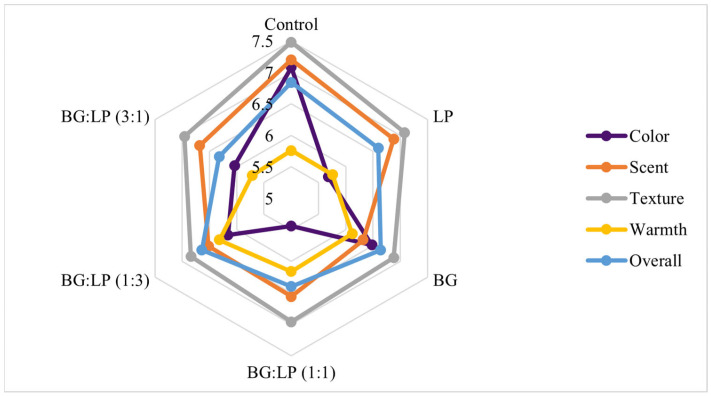
Spider graph of organoleptic results of spice stick balsams containing macroemulsions of lemon pepper (LP), black ginger (BG), and combination.

**Table 1 pharmaceuticals-16-00371-t001:** Active compounds in lemon pepper and black ginger extracts.

Extract	Group	Compound	RT	SI	Area (%)
Lemon pepper	Monoterpenes	D-limonene	11.02	635	0.26
		Verbenol	7.72	671	0.06
		Geraniol	15.72	882	17.26
		Citronellal	13.33	789	1.01
		Geranyl formate	16.67	848	5.63
Black ginger	Phenolics	6-Shogaol	40.36	899	5.73
		6-Iso-shogaol	59.32	634	10.05
	Polyphenols	Gingerol	41.88	686	2.28
	Methylated flavonoids	2-hydroxy-7,3′,4′,5′ tetramethoxyflavone	33.93	586	0.07
	Xanthones	Mangostine	57.19	950	0.33

**Table 2 pharmaceuticals-16-00371-t002:** Characterization of spice stick balsams.

Stick Balsam	Homogeneity	Spread Ability (cm)	Adhesion (s)	pH
Control	Yes	4.90 ± 0.10	70.00 ± 9.54	6.00 ± 0.00
LP	Yes	4.83 ± 0.12	37.67 ± 7.37	5.00 ± 0.00
BG	Yes	4.70 ± 0.10	43.67 ± 1.15	5.00 ± 0.00
BG:LP (1:1)	Yes	4.53 ± 0.06	34.67 ± 3.51	5.00 ± 0.00
BG:LP (1:3)	Yes	4.73 ± 0.15	38.67 ± 5.51	5.00 ± 0.00
BG:LP (3:1)	Yes	4.73 ± 0.06	51.00 ± 3.61	5.00 ± 0.00
Commercial stick balsam (GELIGA^®^)	Yes	4.77 ± 0.06	51.33 ± 4.04	6.00 ± 0.00

Note: LP, lemon pepper; BG, black ginger.

**Table 3 pharmaceuticals-16-00371-t003:** Organoleptic results of spice stick balsam.

Stick Balsam	Color	Scent	Texture	Warmth	Overall
Control	7.08 ± 1.32 ^b^	7.2 ± 1.41 ^a^	7.48 ± 1.23 ^a^	5.76 ± 1.71 ^a^	6.84 ± 1.07 ^a^
LP	5.68 ± 1.44 ^a^	6.88 ± 1.45 ^a^	7.08 ± 1.04 ^a^	5.76 ± 1.54 ^a^	6.6 ± 1.22 ^a^
BG	6.48 ± 1.58 ^ab^	6.32 ± 1.60 ^a^	6.88 ± 1.51 ^a^	6.12 ± 1.36 ^a^	6.64 ± 1.29 ^a^
BG:LP (1:1)	5.44 ± 1.87 ^a^	6.56 ± 1.76 ^a^	6.96 ± 1.46 ^a^	6.16 ± 1.80 ^a^	6.4 ± 1.44 ^a^
BG:LP (1:3)	6.16 ± 1.25 ^ab^	6.52 ± 1.61 ^a^	6.84 ± 1.18 ^a^	6.32 ± 1.38 ^a^	6.64 ± 1.41 ^a^
BG:LP (3:1)	6.04 ± 2.24 ^ab^	6.68 ± 1.38 ^a^	6.96 ± 1.49 ^a^	5.72 ± 1.62 ^a^	6.32 ± 1.46 ^a^

Note: LP, lemon pepper; BG, black ginger. ^a,b^ Means ± SD with different letter in the same column are significantly different at *p* < 0.05 according to stepwise stepdown method.

**Table 4 pharmaceuticals-16-00371-t004:** Macroemulsion formulation.

Composition	Amount (%)
Spice extract *	10
Soy oil	10
Propylene glycol	20
Tween 80	20
Ethanol 96% food grade	15
Aquadest	25

* Spices used were lemon pepper (LP) and black ginger (BG) extracts, giving two macroemulsions.

**Table 5 pharmaceuticals-16-00371-t005:** Spice stick balsam formulation.

Formulation (%)						
Composition	Control (a)	LP (b)	BG (c)	BG:LP (1:1) (d)	BG:LP (1:3) (e)	BG:LP (3:1) (f)
LP *	-	15.00	-	7.50	11.25	3.75
BG *	-	-	15.00	7.50	3.75	11.25
Paraffin	8.00	8.00	8.00	8.00	8.00	8.00
Cera alba	8.00	8.00	8.00	8.00	8.00	8.00
Vaseline	65.00	65.00	65.00	65.00	65.00	65.00
Menthol	3.50	3.50	3.50	3.50	3.50	3.50
BHT	0.50	0.50	0.50	0.50	0.50	0.50

* In macroemulsion form. Note: Control (a); lemon pepper (LP, b); black ginger (BG, c); combination of BG:LP (1:1) (d); combination of BG:LP (1:3) (e); and BG:LP (3:1) (f).

## Data Availability

Data is contained within the article.

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
