# Peer review of "Characterization of Lemon Pepper and Black Ginger Extracts and Macroemulsions as Natural Pain Relievers for Spice Stick Balsam Formulation"

_pharmaceuticals, 2023, doi:10.3390/ph16030371_

Round 1

Reviewer 1 Report

The article by Yanti et al. proposes the use of Lemon Pepper and Black Ginger Extracts as natural pain relievers as a potential alternative to methyl salicylate which is reported to cause allergy and skin irritation. The article is indeed interesting; however, I have some queries:

The abstract is too lengthy. At several points, the authors have explained the methodology up to a level that makes it look like the Material and Methods section. Please make it concise.

Abstract Line NO:9 – Reframe the first sentence. The meaning is not clear.

In Introduction: Line No. 34: What is joint disease? Replace it with some other word. It is misleading.

The authors have started talking about Balsam out of a sudden in abstract as well as in the introduction section. Authors should firstly give a brief description about Balsam. Its composition, prevalence of treatments including Balsam. Is it FDA approved or a part of traditional medicines?

The entire introduction section is very poorly written. It should be rewritten in the light of story that the authors want to present.

            For example: A paragraph highlighting arthritis. Authors should lay stress on its prevalence by including data from authentic sources.

            A paragraph highlighting the prevailing treatments and the limitations of currently prevailing treatments. Why Balsam is preferable over other treatments? What are the limitations associated with the treatments involving Balsam.?

            Next paragraph should highlight the possible solutions? The authors should specifically mention here about their choice of Lemon Pepper and Black Ginger Extracts? Why not others?

Improve the quality of Figure 1.

There is no need to put section 2.1 as a separate section. It can be merged with the subsequent section.

The value of standard error is zero for the yield of LP? What was the number of replicates and how many time the experiments were repeated? How did the authors calculate the yield? It should be included in the Material and Methods’ section?

In line 83 the authors have used the term “strong fragrance” and “fresh fragrance” to describe ethe smell of extracts. What was the methodology and scale used to differentiate the fragrances. Since it is obvious that the lemon and pepper have their own distinct fragrances. How were the authors able to determine the fragrance fresh or strong?

The authors have measured the particle size of emulsion. What’s the correlation between the size of emulsion and its antioxidant activity? Was this assay really required?

  “

Author Response

Jakarta, 02 February 2023

To:

Editor of Pharmaceuticals journal

Dear Sir,

Enclosed please find our revised manuscript entitled: “Characterization of Lemon Pepper and Black Ginger Extracts and Macroemulsions As Natural Pain Relievers for Spice Stick Balsam Formulation”. We have revised it based on the valuable inputs from 3 reviewers. We also attached our responses to all reviewer’s comments and suggestion. We do apologize to revise and resubmit it late.

We hope that this manuscript could be considered to be published in Pharmaceuticals journal. Thank you so much for your kind assistance

Sincerely,

Yanti

Author Response

(The authors gave the same response as above.)

Reviewer 3 Report

- In the opinion of the reviewer, the introduction is too short.

- It is puzzling what guided the determination of the order of the compounds in Figure 1 in Table 1? Shouldn't for example retention time be taken into account?

- All balm stick products had TPC value of 0 CFU/g. According to the reviewer, Table 3 is redundant. Wouldn't it be better to combine this subchapter with another.

- Wrong table number. (Table 6. Organoleptic results of spice stick balm)

- Table 6 and Figure 4 show the same data. One form of presentation of the results should be selected.

- Line 177-178: 'However, when compared with the characteristics of the commercial stick balm, the spice balsams have similar characteristics to the commercial product., - Was a commercial product compared in this study? If a commercial product has not been tested, the literature specifying exemplary parameters of a commercial product should be quoted.

- The methodology does not explain how the control sample was prepared.

- In the discussion of the results, the authors write that the BHT preservative was used. To what extent the lack of presence of microorganisms resulted from the addition of a preservative, and to what extent from the addition of spice extract.

- Conclusion: This part is more of a summary now. There are no clearly formulated conclusions.

Author Response

(The authors gave the same response as above.)

Round 2

Reviewer 1 Report

The authors have incorporated all the suggestions. The script has been improved. I recommend the acceptance.

Reviewer 3 Report

Most of the comments have been taken into account. The explanations and corrections made by the authors can be considered sufficient.